# Lime (*Citrus aurantifolia* (Christm.) Swingle) Essential Oils: Volatile Compounds, Antioxidant Capacity, and Hypolipidemic Effect

**DOI:** 10.3390/foods8090398

**Published:** 2019-09-07

**Authors:** Li-Yun Lin, Cheng-Hung Chuang, Hsin-Chun Chen, Kai-Min Yang

**Affiliations:** 1Department of Food Science and Technology, Hungkuang University, Taichung 433, Taiwan; 2Department of Nutrition, Hungkuang University, Taichung 433, Taiwan; 3Department of Cosmeceutics, China Medical University, Taichung 404, Taiwan; 4Department of Hospitality Management, Mingdao Unicersity, ChangHua 523, Taiwan

**Keywords:** lime essential oil (LEO), antioxidant capacity, hypolipidemic effect

## Abstract

Lime peels are mainly obtained from the byproducts of the juice manufacturing industry, which we obtained and used to extract essential oil (2.3%) in order to examine the antioxidant and hypolipidaemic effects. We identified 60 volatile compounds of lime essential oil (LEO) with GC/MS, of which the predominant constituents were limonene, γ-terpinene, and β-pinene. Lime essential oil was measured according to the DPPH assay and ABTS assay, with IC_50_ values of 2.36 mg/mL and 0.26 mg/mL, respectively. This study also explored the protective effects of LEO against lipid-induced hyperlipidemia in a rat model. Two groups of rats received oral LEO in doses of 0.74 g/100 g and 2.23 g/100 g with their diets. Eight weeks later, we found that the administration of LEO improved the serum total cholesterol, triglyceride, low-density lipoprotein cholesterol, alanine aminotransferase, and aspartate transaminase levels in the hyperlipidemic rats (*p* < 0.05). Simultaneously, the LEO improved the health of the rats in terms of obesity, atherogenic index, and fatty liver.

## 1. Introduction

The genus *Citrus* (Rutaceae) is one of the most commonly consumed and widely distributed fruits in the world. Citrus fruits can be classified into many types including mandarins, tangerines, oranges, pomelos, hybrids, lemons, limes, etc. As one of the most consumed fruits, citrus fruits are also of great economic importance [1]. These fruits are most commonly consumed as fresh whole fruits or juices because of their nutritional values and pleasant flavors [2]. Some of these citrus fruits and juices have also been used as functional foods and drinks with potential application in the treatment of diet-related diseases in people with different health conditions. In this regard, several examples of the use of citrus fruits as therapeutic remedies can be cited, including the use of orange, lime, and lemon juices as remedies for the prevention of kidney stone formation; the use of citrus flavonoids as effective in vivo agents able to modulate hepatic lipid metabolism; the use of orange juice to prevent and modulate inflammatory processes; the use of calamondin and kumquat peel polyphenolics as effective antioxidant agents; the use of grapefruits as agents able to lower blood pressure and to interfere with calcium channel blockers; the use of pomelo for appetizer and stomach tonic; the use of lime juice with honey as as cough reliever [3,4,5,6]. Lime (*Citrus aurantifolia* (Christm.) Swingle) has a smooth, thin skin, is greenish-yellow in color, and has a very small neck, a flat base, and a small nipple at the apex. The highly acidic flesh is juicy with a distinct aroma and flavor [1]. However, lime ripens green to yellow and, hence, tends to become softer to perishable during storage.

Citrus fruits are consumed in substantial quantities worldwide as fresh produce and juice, and most often the peels, which contain a wide variety of secondary components with substantial antioxidant activity in comparison with the other parts of the fruits, are discarded as waste. The global production of citrus fruits has significantly increased during the past few years, having reached 82 million tons in the years 2009–2010, with oranges being the most commercially important citrus fruits, accounting for approximately 50 million of those 82 million tons [7]. In that period, 34% of the oranges produced were used for juice production, with about 44% of the juice oranges used consisting of peel byproducts [7,8]. In other words, an enormous amount of citrus peels are produced every year. Citrus peels, the primary waste product of citrus juice production, are a good source of molasses, pectin, and limonene and are usually dried, mixed with dried pulps, and sold as cattle feed. In general, the wastes and byproducts of citrus fruits contain large amounts of high-added value compounds and show a variety of potentially valuable uses in the technological and health-promoting domains. Amongst the available biologically active compounds in citrus byproducts are polyphenols, carotenoids, and essential oils (EOs) [5,6,7,8].

The polyphenols and carotenoids are known to have numerous health benefits, which are mostly attributed to their antioxidant activity [9,10,11]. Polyphenols have significant potential as a lucrative raw material for the production of functional foods, pharmaceuticals, and cosmetics [10]. A variety of health benefits, such as anti-carcinogenic, anti-mutagenic, anti-allergenic, and antiaging activities have been reported for polyphenols, while their stability during processing has also been studied [7,9]. Citrus essential oils are characterized as mixtures of many components, including terpenes, sesquiterpenes, aldehydes, alcohols, and esters, and can be described as a mixture of terpene hydrocarbons, oxygenated compounds, and non-volatile residues. The volatile chemical compounds of citrus essential oils are among the most distinctive components of citrus fruits for purposes of identifying and evaluating various fruit varieties. d-Limonene, γ-terpinene, linalool, linalyl acetate, α-terpineol, geranyl acetate, terpinolene, and β-pinene are active components of citrus essential oils, and play important roles antioxidant, anti-inflammatory, anticarcinogenic, antidepressant, and antifungal processes [5,6].

Cardiovascular disease (CVD) is an important global public health problem, with hyperlipidemia and hypertension being common related health concerns [12]. Hyperlipidemia is defined as a disorder of lipid metabolism that leads to abnormal increases in triglyceride (TG), total cholesterol (TC), low-density lipoprotein cholesterol (LDL-c), and very low-density lipoprotein cholesterol (VLDL-c) levels, as well as decreases in high-density lipoprotein cholesterol (HDL-c) levels [12,13]. Diet and lifestyle have been found to include various risk factors that are associated with the incidence of CVD. Greater consumption of vegetable and fruit has been found to be associated with lower CVD risk, for example, supplementation with tomato juice produced using a process that increases lycopene’s and dietary fiber’s effects on hyperlipidemia regulation [11,14]. Citrus fruits are a rich source of many flavonoids, especially flavanone glycosides (rutin, hesperidin, hesperetin, and quercetin) and polymethoxyflavones (nobiletin and tangeretin) which make contributions to protection against atherogenesis [1,3,15,16].

Worldwide estimates of lime peel waste is 2.8 million tons/year from citrus processing industries. Clearly, the focus on production and application of essential oils could be beneficially managed and industrialized properly [1]. The present study isolated lime essential oil (LEO) using steam distillation. We then analyzed the chemical constituents of the LEO using gas chromatography/mass spectrometry (GC/MS), while also evaluating the presence of scavenging DPPH and ABTS radicals in the LEO. In addition, we evaluated the hypolipidemic activity of the LEO in rats fed a high-fat diet (containing 0.2% cholesterol and 9.8% lard), observing the lipid composition of the serum, liver, and fecal matter of the rats to evaluate the effects of the LEO on hyperlipidemia rats.

## 2. Materials and Methods

### 2.1. Sample Preparation and Reagents

Limes were harvested from trees at local farms in October 2017. The citrus fruits were separated into edible and inedible portions (peels) and stored at −30 °C before use. 1,1-diphenyl-2-picrylhydrazyl (DPPH), 2,2′-azinobis (3-ethylbenzothiazolin-6-sulfonic acid) (ABTS), cholesterol, and n-alkanes mix solutions were purchased from Merck (Darmstadt, Germany) The TG, TC, LDL-C, and HDL-C were determined using commercial assay enzyme kits from Randox (Crumlin, UK). Cellulose, methionine, choline bitartrate, and other analytical grade chemicals mentioned were purchased from CHEMICAL Co., Ltd. (Miaoli, Taiwan).

### 2.2. Essential Oils Preparation and Analysis

The method followed was previously described by Lin et al. [17]. Lime peel (500 g) was added to 2 L distilled water and homogenized (Waring Blender Model HGB7WTS3, Waring Co., Torrington, CT, USA) for 5 min. Steam distillation was then conducted for 3 h with a slight modification to collect the essential oils. The sample essential oils were dehydrated with anhydrous sodium sulfate and filtered. The essential oils obtained were weighed and the yields were calculated. The products were stored at −20 °C for further use.

The volatile compounds were identified using Agilent 6890 GC equipped with a 60 m × 0.25 mm i.d. DB-1 fused-silica capillary column with a film thickness of 0.25 μm coupled to an Agilent model 5973 N MSD mass spectrometer. The injector temperature was maintained at 250 °C. The GC conditions in the GC/MS analysis were the same as in the GC analysis. The carrier gas flow rate was 1 mL helium/min. The electron energy was 70 eV at 230 °C. The constituents were identified by matching their spectra with those recorded in an MS library (Wiley 7n). Additionally, an n-alkanes (C_5_–C_25_) reference mixture was used to calculate the retention indices (RIs) in relation to those of authentic standards or those in the published literature. The relative content of each compound was calculated as a percent of the total chromatographic area and the results were expressed as means of triplicate experiments.

### 2.3. Antioxidant Capacity Assay

For the purpose of evaluating the antioxidant activity of the small-sized citrus fruit samples, the biochemical methods of total phenolic, total flavonoid, DPPH, and ABTS radical-scavenging assays were used. The tests were carried out in triplicate. The DPPH radical-scavenging activity was detected according to methods used in previous research [17], with different concentrations of LEOs being mixed in a DPPH-radical-containing methanol solution (0.2 mM). After being shaken and incubated for 30 min, each sample was measured by UV absorbance at 517 nm. The scavenging activity of ABTS radicals was modified according to previous research. The ABTS+ were produced by reacting ABTS solution (7 mM) with potassium persulphate (1.4 mM) and allowing the ABTS+ solution to stand in the dark for 16 h. According in previous research [18], the ABTS+ solution was diluted with methanol to a suitable absorbance of 0.80 ± 0.05 at 734 nm. Different concentrations of LEOs were then added to the ABTS+ solution and mixed vigorously. After reacting for 3 min at room temperature, the absorbance at 734 nm was again measured.

### 2.4. Animals and Treatments

Specific pathogen-free (SPF) male Wistar rats (6 weeks old) were purchased from the National Laboratory Animal Center (NLAC), Taipei City, Taiwan. The animals were housed in the animal facility at Hungkuang University at a temperature of 22 ± 1 °C and 50% to 60% relative humidity, with a 12 h light–dark cycle (light on at 7:00 a.m.). Before the experiments, the Wistar rats were acclimatized for 1 week to the environment and diet. The animal experiment was approved by the Ethics Committee for the Care of Animals and Animal Experiments of Hungkuang University (the affidavit indicating approval of the animal protocol used has been attached herewith).

The experimental design is shown in Table 1. A total of 32 Wistar rat were randomly divided into 4 groups for treatment (*n* = 8/per group): (1) the CN group, which was fed an AIN-76A rodent diet; (2) the HF group, which was fed an AIN-76A rodent diet with 0.2% cholesterol and 9.8% lard; (3) the LEO-N group, which was fed an AIN-76A rodent diet with 0.74% LEOs, 0.2% cholesterol, and 9.8% lard; and (4) the LEO-H group, which was fed an AIN-76A rodent diet with 2.23% LEOs, 0.2% cholesterol, and 9.8% lard. The vehicle treatment was the volume of solution to body weight (BW). The food intake and water consumption were monitored daily and BW was recorded weekly.

The experimental period lasted for 12 weeks, during which the body weight and amount of food consumed were recorded every 2 days until the end of the experiment. After 8 weeks of feeding, the rats were fasted for 12 h and CO_2−_anethesized to allow the collection of blood from their abdominal aortas. The blood samples were immediately centrifuged at 3200 rpm for 15 min using a Kubota-3740 centrifuge to separate and collect the sera, which were used later for determination of the AST, ALT, TG, TC, LDL-C, and HDL-C with commercial assay kits (Randox, Crumlin, UK). After the animals were euthanized, their livers were dissected and rinsed with saline; the adsorbed water was sucked off of the surface, and the samples were weighed. Lipids were extracted from the liver and feces of each rat using chloroform–methanol (2:1, *v*/*v*) following to previous research [17]. We then analyzed the TG and TC levels in the liver and the neutral sterols in the feces.

Liver tissues were then carefully removed, minced, and fixed in 10% formalin. All samples were embedded in paraffin and cut into 4 μm thick slices for morphological and pathological evaluations. Each tissue sample was stained with hematoxylin and eosin (H&E) and examined under a light microscope equipped with a CCD camera (BX-51, Olympus, Tokyo, Japan) by a veterinary pathologist.

### 2.5. Statistical Analysis

All data are expressed as mean ± SD. Statistical differences were analyzed by one-way analysis of variance (ANOVA) and the Cochran–Armitage test for trend analysis of the dose–effect of KOT supplementation using SAS 9.0 (SAS Inst., Cary, NC, USA). *p* < 0.05 was considered statistically significant.

## 3. Results and Discussion

Essential oils and oleoresin were found in citrus peels and produced by the cells within the rind, the bulk of which were mainly obtained as byproducts of the juice manufacturing industry [6]. In this study, it was found that the essential oil extraction rate via steam distillation for lime was 2.3% (*w*/*w*). GC/MS was used to identify the volatile components in the extracted oils. We detected 60 volatile compounds in the LEO (Table 2), including 13 monoterpenes (80.36%), 20 sesquiterpenes (6.51%), 3 terpene aldehydes (3.79%), 12 terpene alcohols (4.54%), 8 oxygen-containing aliphatics (0.43%), 3 terpene esters (2.85%), and 1 terpene ketone (0.01%). Of those compounds, the content of limonene was the highest (42.35%), followed by those of γ-terpinene (15.44%), β-pinene (12.57%), α-pinene (3.12%), neryl acetate (2.2%), and sabinene (2.12%). Monoterpene components of LEO could be described as “top note”, because they were sharp and perceived immediately upon application [19,20].

Monoterpene synthases manage variously volatile compounds from citrus essential oils. Simultaneously, the different varieties of citrus, as well as the various geographical locations and climates in which they are produced play important roles in the specific levels and types of the various compounds that they contain [21]. In a previous study, for example, limonene was found to be a major component of the essential oils from two Malaysian citrus species, namely, *Citrus grandis* and *Citrus microcarpa*, accounting for 81.6% and 94% of their respective oil contents [22]. High level limonene does not contribute to flavor and fragrance in essential oil, and it is unstable in the presence of heat and light. The removal of limonene is conducted to increase the concentration of oxygenated flavor compounds and to improve the shelf stability and handling properties in industry [23]. However, LEO had low percentages of limonene compare with other citrus fruits, which can avoid diterpene processing losses.

In the past decade, natural antioxidants have mostly been extracted from plant sources and they have generally been found to be safe [9,16,24,25]. The scavenging of radicals by antioxidants most commonly occurs via two mechanisms, which involve the transfer of either a hydrogen atom or an electron to convert the radical into a stable compound [25]. The DPPH scavenging effect of the LEO investigated in this study were found to range from 10.65–66.44% in 0.08–3.46 mg/mL, with an IC_50_ value of 2.36 mg/mL (Figure 1). The ABTS scavenging effect of the LEO were found to range from 16.62–51.07% in 0.01–0.27 mg/mL, with an IC_50_ value of 0.26 mg/mL (Figure 1). In a previous study, the antioxidant activity of the essential oils from Gannan navel oranges was also measured using a DPPH assay and ABTS assay, and they were found to have IC_50_ values of 2.19 ± 0.20 mg/mL and 2.00 ± 0.19 mg/mL [26].

At the end of the experimental period, the different experimental groups in this study did not show significant differences in terms of feed consumption. The mean body weights of the groups, however, were significantly different (*p* < 0.05), with the LEO-H group (465 g) and LEO-N group (560 g) having lower mean weights than the HF group (606 g). In terms of food efficiency, the LEO-H group (6.36) and LEO-N group (9.87) efficiencies were lower than that of the HF group (11.31) (data not shown). In our previous experiments, it was also shown that essential oils reduce weight gain from high-fat diets and decrease food efficiency [17]. The rats receiving LEO also exhibited lower TC, TG, and LDL-C levels (*p* < 0.05) compared with the HF group. The LEO-H group also exhibited TC levels 28.8% lower, TG levels 39.6% lower, and LDL-C levels 11.2% lower than the HC group (*p* < 0.05).

In recent decades, consumer-increased intake of high simple carbohydrates and high-fat diets has been found to have obvious effects on the serum triglyceride concentrations of people worldwide [13]. Atherogenic index (AI) and HDL-C/TC ratio (HTR) are risk factors for cardiovascular disease. Studies have found that low AI indexes and low plasma triglyceride concentrations can reduce the incidence of cardiovascular disease. In contrast, if the AI is high, the incidence of CVD will increase. The result of AI, CN, HF, LEO-N, and LEO-H were 2.7, 4.1, 2.5, and 2.2, respectively (Table 3.). The hepatic lipid results of the groups were also significantly different (*p* < 0.05), with the LEO-H group (25.7%) and LEO-N group (22.3%) showing decreased percentages as compared to the HF group in terms of TC, and the LEO-H group (27.6%) and LEO-N groups (23.1%) showing decreased percentages as compared to the HF group in terms of TG. About 55% of the cholesterol in the diet will be absorbed by the body. Human cholesterol mediates liver LDL receptor, cholesterol acyltransferase (ACAT), and HMG-CoA reductase in jejunum cells [13,27,28], leading to the following effects: (i) an increased number of LDL receptors results in decreased LDL-C, (ii) the inhibition of HMG-CoA reductase activity decreases cholesterol synthesis in vivo, and (iii) the inhibition of ACAT activity increases the number of LDL receptors [27]. In addition, high-cholesterol diets reduce the ß-oxidation and hinder the metabolism of fatty acids [28]. A large amount of TG is accumulated in the liver to form cholesterol in the liver. Cholesterol is excreted in the form of cholic acid after being digested and absorbed by the human body, while it is excreted in the form of neutral sterols when not absorbed. The amounts of neutral sterols in the fecal samples of the CN, HF, LEO-N, and LEO-H groups in this study were 4.0, 7.6, 8.8, and 10.0 mg/g, respectively. Hepatitis elevated the ALT and AST of serum, and ALT had sensitivity and specificity with liver. In the fatty liver rats in this study, the ALT levels in the serum and liver increased and decreased, respectively [29]. The ALT results in the CN, HF, LEO-N, and LEO-H groups were 32.1, 51.3, 32.5, and 28 u/L, respectively. The AST results in the CN, HF, LEO-N, and LEO-H groups were 87.0, 104.7, 75.6, and 54.3 u/L, respectively. With respect to the histopathological results for the rat livers, we observed fatty infiltration with micro- and macro-vesicles of hepatocytes with multifocal inflammatory cell infiltration in the HF group (Figure 2.). The LEO-N and LEO-H groups exhibited improved fatty infiltration compared to the HF group. In a previous study, EO of Finger Citron also improved fatty infiltration [30].

Lipo-oxidative reactions in humans usually lead to irreversible tissue damage. Lipid peroxides frequently decompose to reactive aldehydes that react with nucleic acids, proteins, and lipids, causing tissue and organ damage [31]. The oxidative modification of lipids, in particular LDL, is thus suggested to play a key role in atherosclerosis, while dietary antioxidants have been shown to prevent LDL oxidation [32]. This research has confirmed that LEO has antioxidant and hypolipidemic effects. The bioactivity of essential oils is determined by non-single compounds. The exploration of a variety of compound essential oils has shown that when the proportion of citral or γ-terpinene is increased, the oils have better antioxidant capacity [33]. Another study showed that γ-terpinene inhibits LDL oxidation [34]. In addition, the cardiovascular activity of essential oils and their individual volatile constituents, such as α-terpineol, α-terpinene, and γ-terpinene, all have potent antioxidative capabilities resulting from the restoration of antioxidant enzyme activities [35]. Limonene, linalool, and 4-terpineol are beneficial in terms of preventing cardiovascular diseases and hypertension. Essential oils and polyphenol had physiological synergistic effects similar to CVD [36].

## 4. Conclusions

Citrus essential oils have been classified as generally recognized as safe (GRAS) material. Used the distilled EO haven’t the furanocoumarins, while the expressed oils carry a low risk of phototoxicity. Essential oils may contribute significantly toward the evolutionary survival of the respective organism, for reproduce in vitro results in the human or animal model. In summary, LEO exhibits potent DPPH and ABTS free radical scavenging activity, being highly effective as inhibitors of lipid peroxidation. The LEO was found to reduce LDL to counteract hyperlipidemia. However, the specific molecular mechanisms by which essential oils induce their effects on hyperlipidemia-induced diseases require further investigation. In the future, LEO may be broadly used as a natural food additive in several food and beverage products. It not only improves sensory evaluations of food but also provides beneficial pharmacological effects to humans.

## Figures and Tables

**Figure 1 foods-08-00398-f001:**
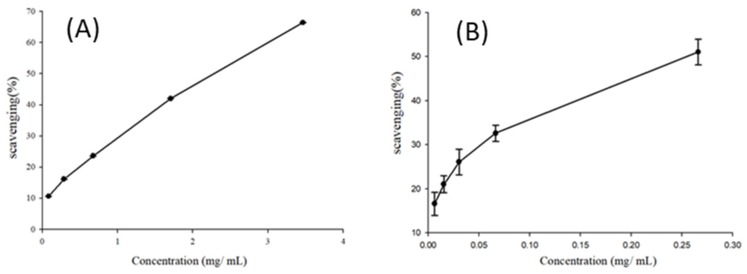
Comparison of EC_50_ values toward 1,1-diphenyl-2-picrylhydrazyl (DPPH) assay (**A**) and 3-ethylbenzothiazolin-6-sulfonic acid (ABTS) (**B**) radical scavenging rates of lime essential oils.

**Figure 2 foods-08-00398-f002:**
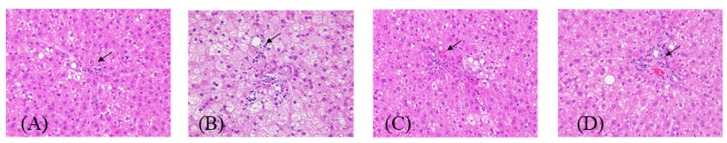
Histopathological changes of lime on liver in Wistar rats induced by high-fat diets. (**A**: Control, **B**: HF, **C**: LEO-N, **D**: LEO-H) H&E stain, 400×. Control (**C**): normal diet, HF: high-fat diet, lime essential oil normal dose (LEO-N): high-fat diet containing 0.744% lime essential oils, lime essential oil normal high dose (LEO-H): high-fat diet containing 2.232% lime essential oils.

**Table 1 foods-08-00398-t001:** The animal feed formula for testing the regulation of hypolipidemic and body fat and the effects of lime on body weight and food efficiency of the high-fat diet of Wistar rat.

Groups	CN	HF	LEO-N	LEO-H
Diet Content (%)				
Casein	20	20	20	20
d,l-methionine	0.3	0.3	0.3	0.3
Corn starch	15	5	5	5
Sucrose	50	50	50	50
Cellulose	5	5	5	5
Ain 76 Mineral	3.5	3.5	3.5	3.5
Ain 76 Vitamin	1	1	1	1
Choline bitartrate	0.2	0.2	0.2	0.2
Corn oil	5	5	5	5
Lard	-	9.8	9.8	9.8
Cholesterol	-	0.2	0.2	0.2
Total	100	100	100	100
Lime essential Oil	-	-	0.744	2.232

Control (CN): normal diet, HF: high-fat diet, lime essential oil normal dose (LEO-N): high-fat diet containing 0.744% lime essential oils, lime essential oil normal high dose (LEO-H): high-fat diet containing 2.232% lime essential oils.

**Table 2 foods-08-00398-t002:** Composition of essential oil of lime peel.

Compounds *	RIs ^#^	Composition (%)
Monoterpenes		
α-thujene	925	1.01 ± 0.15
α-pinene	941	3.12 ± 0.26
camphene	950	0.14 ± 0.01
sabinene	967	2.12 ± 0.17
β-pinene	971	12.57 ± 0.8
β-myrcene	983	1.92 ± 0.03
α-phellandrene	996	0.13 ± 0.01
α-terpinene	1009	0.37 ± 0.02
o-cymene	1013	1.30 ± 0.21
limonene	1030	42.35 ± 1.89
β-ocimene	1040	0.28 ± 0.07
γ-terpinene	1051	15.44 ± 1.89
α-terpinolene	1094	1.12 ± 0.06
Terpene alcohols		
linalool	1087	0.64 ± 0.03
fenchol	1107	0.04 ± 0.00
1-terpinenol	1125	0.04 ± 0.00
borneol	1156	0.06 ± 0.04
4-terpineol	1164	0.40 ± 0.57
α-terpineol	1174	1.58 ± 0.11
nerol	1203	0.85 ± 0.15
geraniol	1311	0.58 ± 0.07
α-bisabolol	1615	0.01 ± 0.01
β-santalol	1642	0.08 ± 0.06
ledol	1656	0.08 ± 0.06
α-bisabolol	1669	0.18 ± 0.02
Terpene aldehydes		
β-citronellal	1131	0.05 ± 0.00
α-citral	1197	1.98 ± 0.27
β-citral	1233	1.76 ± 0.25
Terpene ketones		
camphor	1118	0.01 ± 0.01
Terpene esters		
citronellyl acetate	1328	0.05 ± 0.03
neryl acetate	1343	2.20 ± 0.23
trans-geranyl acetate	1357	0.60 ± 0.10
Sesquiterpenes		
δ-elemene	1331	0.23 ± 0.01
α-farnesene	1373	0.41 ± 0.02
β-elemene	1382	0.31 ± 0.02
trans-α-bergamotene	1408	0.12 ± 0.01
γ-elemene	1427	0.05 ± 0.01
β-caryophyllene	1430	0.86 ± 0.21
trans-α-bergamotene	1432	1.44 ± 0.09
trans-β-farnesene	1442	0.16 ± 0.02
α-humulene	1448	0.11 ± 0.03
β-santalene	1454	0.07 ± 0.00
γ-curcumene	1470	0.05 ± 0.01
germacrene-D	1473	0.23 ± 0.02
cis-α-bisabolene	1489	0.18 ± 0.01
β-selinene	1492	0.02 ± 0.02
β-bisabolene	1499	2.00 ± 0.13
cis-γ-bisabolene	1505	0.06 ± 0.00
trans-γ-bisabolene	1520	0.02 ± 0.00
trans-α-bisabolene	1529	0.07 ± 0.00
germacrene-B	1554	0.10 ± 0.02
α-santalene	1598	0.02 ± 0.02
Oxygen-containing aliphatics		
p-menth-2-en-1-ol	1108	0.05 ± 0.00
camphene hydrate	1136	0.01 ± 0.00
isopulegone	1159	0.02 ± 0.03
decanal	1181	0.17 ± 0.01
bornyl acetate	1269	0.02 ± 0.00
undecanal	1284	0.03 ± 0.01
tetradecanal	1388	0.12 ± 0.02
dodecanal	1395	0.01 ± 0.01

Each value is the mean of three replications; ^#^ RI: retention index; * identified via comparison of the mass spectra with the RI.

**Table 3 foods-08-00398-t003:** Effects of lime essential oils on serum lipids, cholesterol, and triglyceride in the liver and neutral sterols in the feces of Wistar rats induced by high-fat diets.

Groups	CN	HF	LEO-N	LEO-H
Serum				
AST (U/L)	87.0 ± 8.0 ^a^	104.7 ± 8.8 ^a^	75.6 ± 8.7 ^ab^	54.3 ± 6.0 ^b^
ALT (U/L)	32.1 ± 2.8 ^b^	51.3 ± 5.0 ^a^	32.5± 5.8 ^b^	28.0 ± 4.7 ^b^
TC (mg/dL)	40.2 ± 6.3 ^b^	49.9 ± 4.9 ^a^	36.6 ± 3.7 ^b^	35.5 ± 6.3 ^b^
TG (mg/dL)	59.6 ± 12.8 ^b^	72.6 ± 14.4 ^a^	50.8 ± 12.5 ^b^	43.8 ± 6.9 ^c^
LDL (mg/dL)	32.8 ± 3.4 ^a^	39.2 ± 1.4 ^b^	31.5 ± 2.9 ^a^	34.8 ± 3.9 ^a^
HDL (mg/dL)	10.7 ± 1.5 ^a^	9.8 ± 0.9 ^a^	10.4 ± 1.5 ^a^	11.1 ± 2.6 ^a^
HTR (%)	26.7 ± 0.5 ^a^	19.63 ± 0.5 ^b^	28.3 ± 1.2 ^a^	30.9 ± 1.8 ^a^
AI	2.7 ± 0.1 ^b^	4.1 ± 0.1 ^a^	2.5 ± 0.2 ^b^	2.2 ± 0.2 ^b^
Liver				
TC (mg/dL)	58.4 ± 1.2 ^b^	72.5 ± 2.8 ^a^	56.3 ± 2.6 ^b^	53.8 ± 3.1 ^b^
TG (mg/dL)	57.0 ± 2.3 ^b^	77.3 ± 1.5 ^a^	59.4 ± 1.8 ^b^	55.9 ± 2.2 ^b^
Feces				
Neutral sterols (mg/g)	4.0 ± 0.4 ^b^	7.6 ± 1.5 ^b^	8.8 ± 0.4 ^b^	10.0 ± 0.4 ^a^

All values are mean ± SD, and the values in a column not sharing a common superscript letter are significantly different from one another according to Duncan’s range test (*p* < 0.05) (*n* = 8). HTR = (HDL-Cholesterol/total Cholesterol) × 100; AI, atherogenic index = (total Cholesterol—HDL-Cholesterol)/HDL-Cholesterol). Control (CN): normal diet, HF: high-fat diet, lime essential oil normal dose (LEO-N): high-fat diet containing 0.744% lime essential oils, lime essential oil normal high dose (LEO-H): high-fat diet containing 2.232% lime essential oils.

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
