# Peer review of "Lime (Citrus aurantifolia (Christm.) Swingle) Essential Oils: Volatile Compounds, Antioxidant Capacity, and Hypolipidemic Effect"

_foods, 2019, doi:10.3390/foods8090398_

Round 1

Reviewer 1 Report

The experiment and results obtained are worth and useful, but the way of their presentation here is not acceptable. Therefore, I suggest to the authors to pay much more attention to some very important points and eventually resubmit the paper in a different form.

I am highlighting here some very big lacks the authors should work on.

-- Some general comments:

- The discussion part (which should be of the greatest importance) is given poorly, and does not include the review of available literature data. In respect to that fact, first of all I suggest to add references and discuss all the previous data that could be find in literature.

- It is completely unacceptable use abbreviations in a random way, especially when it comes to the abstract where they should not be used at all. ''LEO'' has not been explained but it should be in its first appearance in the text.

- Using LEOs, as a plural form, through the whole text should be reconsidered (the explanation is in the last part of this review report).

- In the title: please add ''activity'' or ''efficacy'' after hypolipidemic.

-- Comments by parts:

1. Introduction

- Line 28: Citrus reticulata (it is not reticulatE but reticulatA) has been specified as the main representative of this group of plants (Citrus species): for which reason?

- Lines 41-44: Species description or lime fruit description? The authors should specify this, and I strongly suggest to give the species description as a part of the Introduction.

- Lines 45-71: The whole paragraph is full of general information given as sentences not related so well (for instance, the monoterpene synthesis as very general information): this whole part should be re-written. 

2. Materials and Methods

- Line 98: Which chemicals? All material used should be specified in this paragraph. 

- The method for EO isolation should be precisely described; also, a relevant reference (or references) should be cited here.

- The other methods used should be described (for instance, TPC and TFC) along with relevant references (for the most of methods, adequate references are missing).

3. Results and Discussion

- Lines 168-172: General information that should not be mentioned here. 

- It is not clear from the results (Table 2) how many samples of EO have been analysed and in which way: is it 3 different EOs (different, meaning obtained from different plant material) or 3 EOs obtained from 3 separate extraction procedures (but from the same plant material; in that case it can not be named different EOs but the samples A, B and C)? It is very important to highlight that fact particularly taking into account the constant appearance of the abbreviation LEOs (plural) which seem to be only 1 EO sample.

- Lines 174-179: I would avoid using term ''kinds'' in this content.

- Line 184: Citrus varieties? Here the authors should pay much more attention to the taxonomic status and do not mix different taxonomic categories (for instance, as the explanation in the line 187 the authors used ''species'').

- Lines 183-195: The whole paragraph is not necessary since it is full of very general information (this is the discussion part, not the introduction).

- Lines 196-206: Different concentrations/dilutions are in question, not different EO samples.

- Line 230: Please, re-write the following sentence and make it much more precise: the result of A.I., CN, HF, LEO-N and LEO-H were 2.7, 4.1, 2.5 230 and 2.2.

- Lines 258-277: The whole paragraph is not necessary since it is full of very general information (this is the discussion part, not the introduction).

- The discussion part is given quite poorly, and does not include the review of available literature - actually, that part is completely missing and it is very important to be added/considered to increase the paper's value.

4. Conclusions

As it has been stated in the title itself, it is plural form of all the data presented in the paper that should be shortly mentioned again as the main highlights. However, the authors have not paid attention to this. Thus, I recommend to revise the conclusion part: this part should be expanded and clearly presented to a reader. It is given poorly - just the main conclusions obtained from the experiments done. Future perspectives and recommendations of potential uses are missing.

Author Response

August 15 2019

Editor-in-Chief

Foods

Dear Editor

Thank you for considering the revised version of our manuscript: foods-565684 "Lime (Citrus aurantifolia (Christm.) Swingle) essential oils: volatile compounds, antioxidant capacity and hypolipidaemic", by Lin. et al. for publication in Foods. We are thankful to the referees and the Editor for pointing out some important modifications needed in the report. We believe that the comments have been highly constructive and very useful to restructure the manuscript. We have thoughtfully taken into account these comments. The explanation of what we have changed in response to the reviewers’ concerns is given point by point in the following pages. The manuscript has been revised as suggested. Revised portions have been marked with yellow bottom.

Referees Comments

The reviewer’s comment-Introductionl

- Line 28: Citrus reticulata (it is not reticulatE but reticulatA) has been specified as the main representative of this group of plants (Citrus species): for which reason?

- Lines 41-44: Species description or lime fruit description? The authors should specify this, and I strongly suggest to give the species description as a part of the Introduction.

- Lines 45-71: The whole paragraph is full of general information given as sentences not related so well (for instance, the monoterpene synthesis as very general information): this whole part should be re-written. The authors’ Answer:

I have reconfirmed and the revised portions are shown in line 26 (The genus Citrus (Rutaceae), line 42-43 (Add a new description) and line 67-70, line 76-82, line 83-85 (re-written).

The reviewer’s comment-Materials and Methods

- Line 98: Which chemicals? All material used should be specified in this paragraph.

- The method for EO isolation should be precisely described; also, a relevant reference (or references) should be cited here.

- The other methods used should be described (for instance, TPC and TFC) along with relevant references (for the most of methods, adequate references are missing)

The authors’ Answer:

I have reconfirmed and the revised portions are shown in line 96-98 (Add a new material), line 102, line 122 and line 127 (Add references)

The reviewer’s comment-Results and Discussion

Q1.

- Lines 174-179: I would avoid using term ''kinds'' in this content.

- Line 230: Please, re-write the following sentence and make it much more precise: the result of A.I., CN, HF, LEO-N and LEO-H were 2.7, 4.1, 2.5 230 and 2.2.

Q2.

- It is not clear from the results (Table 2) how many samples of EO have been analysed and in which way: is it 3 different EOs (different, meaning obtained from different plant material) or 3 EOs obtained from 3 separate extraction procedures (but from the same plant material; in that case it can not be named different EOs but the samples A, B and C)? It is very important to highlight that fact particularly taking into account the constant appearance of the abbreviation LEOs (plural) which seem to be only 1 EO sample.

- Line 184: Citrus varieties? Here the authors should pay much more attention to the taxonomic status and do not mix different taxonomic categories (for instance, as the explanation in the line 187 the authors used ''species'').

- Lines 196-206: Different concentrations/dilutions are in question, not different EO samples.

Q3.

- Lines 183-195: The whole paragraph is not necessary since it is full of very general information (this is the discussion part, not the introduction).

- Lines 168-172: General information that should not be mentioned here.

- Lines 258-277: The whole paragraph is not necessary since it is full of very general information (this is the discussion part, not the introduction).

- The discussion part is given quite poorly, and does not include the review of available literature - actually, that part is completely missing and it is very important to be added/considered to increase the paper's value. The authors’ Answer:

A1. I have reconfirmed and the revised portions.

A2. I have reconfirmed and the revised portions. We unified the LEO in the manuscript, and the abbreviation table was added to the manuscript.

A3. I have reconfirmed and the revised portions are shown in line 171-172, line 188-193, line 255-268

The reviewer’s comment- Conclusions

As it has been stated in the title itself, it is plural form of all the data presented in the paper that should be shortly mentioned again as the main highlights. However, the authors have not paid attention to this. Thus, I recommend to revise the conclusion part: this part should be expanded and clearly presented to a reader. It is given poorly - just the main conclusions obtained from the experiments done. Future perspectives and recommendations of potential uses are missing.

The authors’ Answer:

I have reconfirmed and the revised portions are shown in line 270-277

Many typographical errors and abbreviations have been revised. All the lines and pages indicated above are in the revised manuscript. We hope that all these changes fulfill the requirements to make the manuscript acceptable for publication in foods.

Looking forward to hearing from you soon.

Sincerely yours,

Kai-Min Yang

Reviewer 2 Report

The authors present a manuscript entitled “Lime (Citrus aurantifolia (Christm.) Swingle) Essential Oils: Volatile Compounds, Antioxidant Capacity and Hypolipidaemic”. The topic is very topical and the contribution proposed by the authors is certainly interesting. In my opinion, the work presents enough innovative ideas and is therefore worthy of being published on FOODS. In order to improve the quality of the manuscript, I suggest to make only some variations, as follows:

Line 59-60: the authors could insert some references concerning the importance of carotenoids and polyphenols. Here are some suggestions:

1) Journal of Food Composition and Analysis. (2016), 53, Pages 61-68.

doi.org/10.1016/j.jfca.2016.08.008

2) European Food Research and Technology. (2008) 226, 327. doi.org/10.1007/s00217-006-0541-4

Ln 80-84: the information reported does not seem to be of interest for the job and in any case not well correlated with the aims of the work. I suggest to eliminate them and focus attention on the bibliography concerning the biological value of the bioactive molecules present in the considered matrix.

Ln 90: It might be useful to insert here a sentence that highlights the innovative value of the work.

Ln 101-103: please, provide more details about the extraction method, the apparatus used etc., or alternatively, if you have used a method from literature, then provide the reference.

Ln 124: reference [18] should be moved at line 122 (“…according to previous research”).

Ln 156: please, provide more details about the lipid extraction (volume, time, technique). Chloroform/methanol 2:1 followed the Folch protocol? If this is the case, please add the reference

Ln 194-195: please, rephrase the following sentences more clearly: “the 5, 10 and 20-fold condensation of essential oil for orange by deterpene, which limoene were 72.08, 73.77 and 23.24 % [22]. However, the LEOs have low percentages of limonene, which can avoid deterpene processing losses”.

Ln 196: please add some references about the importance of these biologically active compounds and the sources from which they can be derived. Here are some suggestions:

Food Research International. (2018), 111, Pages 291-298. doi.org/10.1016/j.foodres.2018.05.047 J AOAC Int. 2019 Jun 14. doi: 10.5740/jaoacint.19-0128.

Author Response

August 15 2019

Editor-in-Chief

Foods

Dear Editor

Thank you for considering the revised version of our manuscript: foods-565684 "Lime (Citrus aurantifolia (Christm.) Swingle) essential oils: volatile compounds, antioxidant capacity and hypolipidaemic", by Lin. et al. for publication in Foods. We are thankful to the referees and the Editor for pointing out some important modifications needed in the report. We believe that the comments have been highly constructive and very useful to restructure the manuscript. We have thoughtfully taken into account these comments. The explanation of what we have changed in response to the reviewers’ concerns is given point by point in the following pages. The manuscript has been revised as suggested. Revised portions have been marked with yellow bottom.

Referees Comments

The reviewer’s comment

Authors could insert some references

The authors’ Answer:

We inserted some references in the manuscript

Dosoky, N.; Setzer, W. Biological activities and safety of Citrus spp. essential oils. Int J Mol Sci. 2018, 19(7), 1966. Fattore, M.; Montesano, D.; Pagano, E.; Teta, R.; Borrelli, F.; Mangoni, A.; Albrizio, S. Carotenoid and flavonoid profile and antioxidant activity in “Pomodorino Vesuviano” tomatoes. J Food Compost Anal. 2016, 53, 61-68. Montesano, D.; Fallarino, F.; Cossignani, L.; Bosi, A.; Simonetti, M.S.; Puccetti, P.; Damiani, P. Innovative extraction procedure for obtaining high pure lycopene from tomato. Eur Food Res Technol. 2008, 226(3), 327. Delgado, A.M.; Issaoui, M.; Chammem, N. Analysis of Main and Healthy Phenolic Compounds in Foods. J AOAC Int. 2019, Zhu, Z.; Li, S.; He, J.; Thirumdas, R.; Montesano, D.; Barba, F.J. Enzyme-assisted extraction of polyphenol from edible lotus (Nelumbo nucifera) rhizome knot: Ultra-filtration performance and HPLC-MS2 profile. Food Res Int. 2018, 111, 291-298. Cheng, M.C.; Ker, Y.B.; Yu, T.H.; Lin, L.Y.; Peng, R.Y.; Peng, C.H. Chemical synthesis of 9 (Z)-octadecenamide and its hypolipidemic effect: a bioactive agent found in the essential oil of mountain celery seeds. J. Agr. Food Chem. 2010, 58(3), 1502-1508.

The reviewer’s comment

Ln 80-84: the information reported does not seem to be of interest for the job and in any case not well correlated with the aims of the work. I suggest to eliminate them and focus attention on the bibliography concerning the biological value of the bioactive molecules present in the considered matrix.

The authors’ Answer:

I have reconfirmed and the revised portions are shown in line 76-82.

The reviewer’s comment

Ln 101-103: please, provide more details about the extraction method, the apparatus used etc., or alternatively, if you have used a method from literature, then provide the reference.

Ln 156: please, provide more details about the lipid extraction (volume, time, technique). Chloroform/methanol 2:1 followed the Folch protocol? If this is the case, please add the reference

The authors’ Answer:

I have reconfirmed and the revised portions are shown in line 102 and line 158.

The reviewer’s comment

It might be useful to insert here a sentence that highlights the innovative value of the work.

The authors’ Answer:

I have reconfirmed and the revised portions are shown in line 83-85 and line 270-277.

Many typographical errors and abbreviations have been revised. All the lines and pages indicated above are in the revised manuscript. We hope that all these changes fulfill the requirements to make the manuscript acceptable for publication in foods.

Looking forward to hearing from you soon.

Sincerely yours,

Kai-Min Yang

Round 2

Reviewer 1 Report

The discussion and conclusion parts are still missing. Discussing the results obtained means comparing them with the available data in literature - thus, authors should pay attention to that, and putting some references into the text already present, is not an appropriate solution. 

Author Response

I have reconfirmed and the revised portions are shown in line 179-180, line 226-228, line 253-254 and line277-278.
